# "What you say and how you say it" matters: An experimental evidence of the role of synchronicity, modality, and message valence during smartphone-mediated communication

Serena Petrocchi ⓘ *, Laura Marciano, Anna Maria Annoni, Anne-Linda Camerini

Faculty of Communication, Culture and Society, Università della Svizzera italiana, Lugano, Switzerland

* serena.petrocchi@usi.ch

## Abstract

Nowadays, smartphone-Mediated Communication (SMC) has become a popular form of social interactions. The present experimental study manipulated three aspects of messaging in a WhatsApp chat as a form of SMC: synchronicity (immediate vs. time-lagged response), modality (with or without emojis), and valence (empathic accurate vs. empathic inaccurate response). The aim of this study was to investigate whether these three aspects had an impact on perceived social support, interpersonal trust, and personality attribution of the communication partner. The partial mediation of perceived social presence (the evaluation of the communication partner's accessibility) and subjective social presence (the perception of being concordant with him/her) was also examined. Participants were 160 young adults, balanced in gender. They were randomly assigned to different the experimental conditions where they engaged in a manipulated WhatsApp chat with a fictitious same-gender communication partner. Post-questionnaire data were analyzed using Structural Equation Modeling. Message valence (empathic accurate response) and modality (with emojis) significantly predicted higher levels of both forms of social presence. Synchronicity (immediate response) predicted higher levels of perceived but not subjective social presence. Social presence, in turn, was positively associated with social support, while subjective, but not perceived social presence, was positively associated with personality attribution. Neither perceived nor subjective social presence were related to interpersonal trust. Our results show that both what is said and how it is said impact the experience of interpersonal relations in SMC.

## Introduction

Nowadays, the smartphone has become a ubiquitous device in our daily lives, especially among younger populations. In Switzerland, 79% of the population owned a smartphone in 2020. In 2021, the estimated number of smartphone users in the country will reach 7.33 million. Instant messaging and social networking show strong growth as well. In 2019, 2.95 billion

---

**Data Availability Statement:** All relevant data are within the manuscript and its Supporting Information files.

---

**Funding:** ALC and SP received financial support for this work by the Swiss National Foundation under the Grant Number 10DL1C_183199. The funder has no role in the study design, data collection and analysis, decision to publish, or preparation of the manuscript.

**Competing interests:** no authors have competing interests.

people worldwide used smartphone-messaging apps, 2.15 million more compared to the previous year [1].

Although smartphones offer a wide range of functionalities, instant messaging remains among the most popular and used ones [2]. Due to the possibility to use text-based conversations complemented by visual elements such as emojis or images, instant messaging offers a particularly rich way of one-to-one or group communication. However, the dislocation of time and space during the conversation remains a different important characteristic compared to face-to-face communication. Thus, it is not surprising that Smartphone-Mediated Communication (SMC) has received attention by researchers aiming to understand how this form of interaction shapes human relationships (e.g., [3–7]).

Although past research has investigated the differential effects of online and offline communication on psychosocial functioning [8–10], SMC is a fast-changing environment in which technological advancements spread quickly and widely in the population. Nowadays, Internet access on the mobile phone is overtaking home broadband services [11]. Alongside free-to-use instant messenger services such as WhatsApp or Telegram are surpassing mobile SMS and MMS services [12], potentially contributing to the evolution of interpersonal relationships.

The portability of the smartphone, the perception of its indispensability for everyday activities [13], and its predominant use in the context of social interactions and interpersonal communication [14] make smartphone-mediated communication (SMC) and its effect on interpersonal functioning a crucial point to be studied. The meta-analysis by Liu and Chang [15] summarizes the differences in interpersonal closeness across different communication channels, including SMC and instant messaging. Results show that mobile phone-based channels are predominantly used to stay in contact with close friends and that instant messaging is related to friendship quality [16, 17]. What is not yet well known is what are the characteristics of SMC that influence and transform the subjective experience during mediated interactions. This experimental study aims to contribute to scientific evidence on the role of specific aspects of SMC (i.e. synchronicity, modality, and message valence) in the perception of the communication partner and the quality of the interpersonal relationship in the context of SMC. In particular, we focused on the construct of social presence to capture how a communication partner is perceived during an interaction. The quality of the interpersonal relationship has been explicated in terms of perceived interpersonal trust, social support, and attribution of personality characteristics to the communication partner. Before introducing the study, the scientific literature on SMC and related concepts will be reviewed to derive a theoretical model guiding the experimental study and the interpretation of its results.

## Smartphone-mediated communication and social presence

Compared to face-to-face interactions, SMC has specific features that have an impact on the perception of social presence, which, in turn, may influence the evaluation of the other (e.g., personality attribution) and the quality of the interaction itself [18]. First, SMC is characterized by the dislocation of time and space, since it allows to interact both in a synchronous and asynchronous way, in a constant-connected environment [19]. However, little is known about how synchronicity affects the individual's perception of the quality of the social interaction *via* SMC.

A second characteristic of the SMC is its lack of physical, visual, and paralinguistic cues derived from body language and speech intonation. However, principles of Media Richness Theory indicate that the possibility to use non- and para-verbal cues, such as emojis or images, can enhance the message value and mimic the natural language of the communicator [20–22]. The content of the message itself is not a specific feature of the SMC, neither is the valence of

the message (positive or negative). However, both could have a potential impact on the perception of the quality of the social interaction *via* SMC, especially if considered in combination with synchronicity and the use of non- and para-verbal cues.

In SMC, one crucial aspect is how people perceive the other person in the interaction as being "real" or as being "here" [23, 24]. When communication happens through digital devices, this sensation is associated with perceived social presence in human relationships [7, 25]. According to Biocca, Harms, & Burgoon [26], the concept of social presence can be divided into three dimensions. Co-presence, defined as an automatic state, consists in the perception of a coherent source of inputs from an interlocutor classified as an agent [26]. Perceived social presence of the other, involving an interpretative process of the psychological and behavioral engagement, leads to the evaluation of the other's accessibility, i.e. how distant the other is perceived [26]. Finally, subjective social presence is the perception of being concordant with the other, resulting in a feeling of connectedness and symmetry within the relationship [26]. All three dimensions are interconnected and activated in the mediated communication process. To note, perceptions of social presence largely depend on the synchronicity of the interaction, as long pauses in the mediated communication process decrease perceptions of "being there" as well as the awareness and imagination of the other person [27]. Thus, our first hypothesis was:

*HP1*: *Synchronous (vs. asynchronous) SMC increases perceptions of social presence, both at the perceptual and subjective level.*

Furthermore, modality (e.g., the use of verbal and visual cues) determines the experience of social presence [7]. For example, non-verbal cues such as emoticons and emojis can compensate the lack of social context signs and enhance the emotional value of the message [22, 28], leading to higher levels of social presence [29, 30]. In fact, emoticons, originally created using standard text characters [31], are surrogates for non-verbal emotional expression [32]. Nowadays, they are represented by small icons, called emojis, such as smiley or sad faces, embedded in the text message. Emojis are considered the evolution of emoticons, as more expressive and more capable of conveying emotional messages [33]. They strengthen the content of the message [34] and disambiguate the communicational intent, providing information about the personality of the interlocutor [35]. Further advancements can be witnessed in Apple's introduction of memoj stickers for the iPhone in autumn 2019. These stickers are customizable animojis for the users, who can define face shape, skin, eyes color, eyewear, and hairstyle to create an animated representation of themselves. According to Derks, Fischer, & Bos [36], thanks to such cues, the differences between the online communication of emotions compared to the face-to-face one almost disappear. Hence, we hypothesized that:

*HP2*: *Text messages with emojis (vs. text only) in SMC increase perceptions of social presence, both at the perceptual and subjective level.*

Aside from synchronicity and modality, as previously mentioned, the valence of the message content needs to be considered. Social presence comes with the expression of emotions, feelings, and mood [37], which contribute to the psychological involvement of the interlocutor [26]. One element, which has been found to be relevant in the experience of social presence, is the empathic accuracy of the interlocutor [38–40]. Empathic accuracy is defined as the degree to which a person infers what other people think and feel [41, 42], enhancing the experience of positive or negative feelings and emotions. In this vein, an empathic accurate answer to a request is a message in which the sender takes into consideration the receiver's mental state

(i.e., what he/she thinks and how he/she feels) and replies accordingly, thus creating a positive experience. Indeed, empathic accuracy plays a vital role in relationship satisfaction [43] and conflict resolution [44]. In turn, social presence is affected by the emotional responses generated by the mediated communication, and it is triggered by empathic or aversive reactions to the other [26, 45]. Accordingly, we hypothesized that:

> HP3: *Messages with positive valence, i.e. empathic accuracy (vs. negative valence, i.e. empathic inaccuracy) increase perceptions of social presence, both at the perceptual and subjective level.*

## Social presence and perceived trust, social support, and personality of the communication partner

Due to the ongoing digitalization process, trust has been in the focus of researchers and the general public alike as in times of fake news and online grooming (to name two examples) people are more and more concerned with the trustworthiness of online sources and information [46]. Trust can be defined as the expectation held by an individual that another person maintains their words, promises, verbal or written statements [47]. Over time, trust becomes a stable personality characteristic [47, 48]. At the same time, trust is a state characteristic depending on the target person and the content of the communication [49–51]. Trust can equally emerge during online interactions [50, 52–54]. In this vein, social presence has been reported to increase online trust [55, 56]; hence, in online and collaborative contexts [57], social presence has been considered an indispensable prerequisite for the development of interpersonal trust [58]. Additionally, also empathic accuracy (vs. empathic inaccuracy) has been found to have an influence on online trust towards another person [59].

Trust is an important ingredient for the development and maintenance of well-functioning relationships [60–62]. It is tightly linked to social support, defined as the propensity to perceive the other person in a relationship as supportive, reliable, and able to give help and guidance [63]. In particular, being able to establish a connection with the other in a collaborative environment based on empathic responses [39] allows to build a perception of social support and intimacy [64].

Another aspect affected by social presence is the personality attribution to the conversation partner during SMC. Similar to face-to-face evaluations, text-based and online communications arise the formation of first impressions about others' personalities [65], reflecting their online behaviors [67, 68]. Adults use personality traits to describe themselves and others [69], and to interpret the others' behaviors, based on attributional schemes, with different levels of accuracy or attributional styles [70]. The development of the perception of others' personality (e.g., impression formation), is a process involving both automatic and conscious components [71], also based on brief interactions and nonverbal behaviors [72, 73]. The impression formation, however, is limited, context-dependent and highly vulnerable to patterns and biases [74], especially in the earliest stages of an interaction [75]. In particular, people generally tend to rely on specific attributes, which can be summarized by the interpersonal traits of agreeableness, neuroticism and extraversion [76, 77], which are the most used during the personality attribution process [78]. Hence, we state our last hypothesis as follows:

> HP4: *Social presence partially mediates the relations between SMC and interpersonal trust, social support, and personality attribution.*

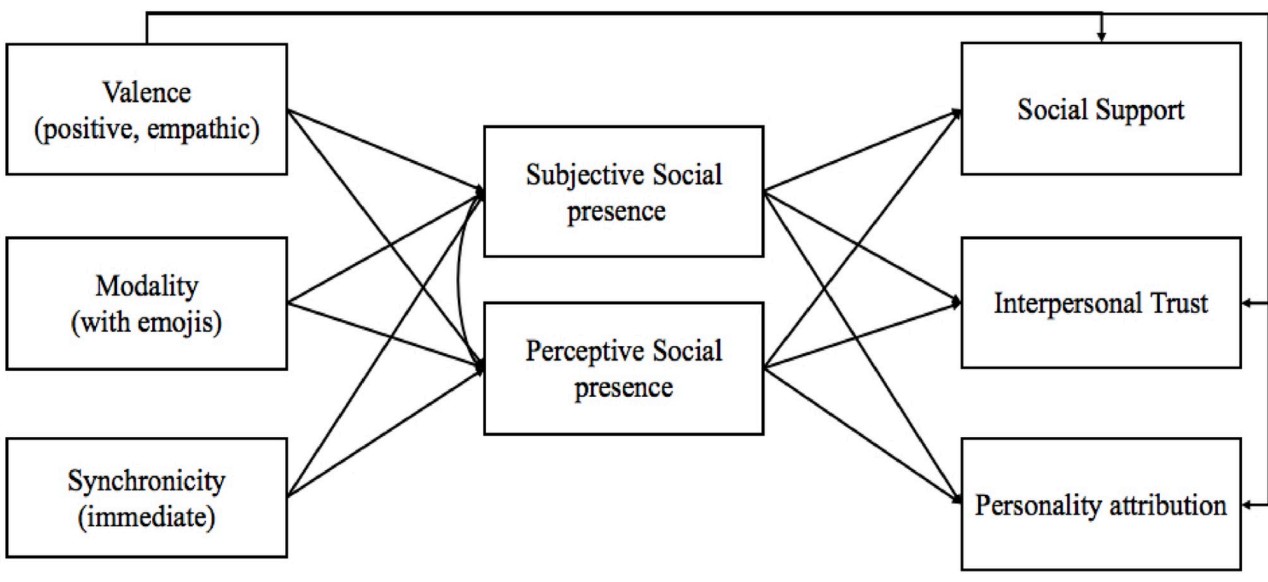

**Fig 1. Theoretical model.** All relationships are hypothesized to be positive.

### Study objectives

Using an experimental design, this study aims to extend the scientific evidence considering the role of specific aspects of SMC in the perception of the communication partner, the conversation, and the interpersonal relationship in general. In particular, three elements of the SMC were manipulated, i.e. synchronicity, modality, and message valence, to investigate whether they have an impact on interpersonal trust, social support, and personality attribution, through the mediation of social presence. Considered previous findings from Park et al. [7], we manipulated SMC by introducing synchronicity delays (high vs. low) in the communication process, and visual elements (text vs. text and emojis). Furthermore, inspired by the textual stimuli introduced by Feng et al. [59] in a study on empathic accuracy and interpersonal trust, we manipulated the valence of the message (positive empathic accurate vs. negative empathic inaccurate). Fig 1 shows the theoretical model tested.

## Materials and methods

### Experimental study design

An experimental study, conducted in a laboratory setting, was used to manipulate one-to-one SMC interaction. Participants were presented with a specific imaginary scenario, in which they just moved in a new city where they did not yet know anyone except for a colleague, equal in age and gender. They got the phone number of this colleague to ask his/her help for some favors they needed. The scenarios were gender adapted. Female participants communicated with a peer called Sophie, while male participants communicated with a peer called Mark. The decision to use same-gender interactions was based on two reasons: First, there is evidence in the literature showing that females and males show different communication styles during mediated communication (e.g., [79]) and that they have different aims during their smartphone interactions (e.g., [80]). To avoid any possible influence of gender, we decided to match the gender of study participants and the fictitious peer. Second, we wanted to avoid any romantic connotations. Dependent on the experimental condition, the peer replied either with an empathic accurate or an empathic inaccurate message (i.e., positive vs. negative valence),

with or without emojis (i.e., high vs. low richness of modality), and immediately after the participant sent the request or 7 minutes later (i.e., synchronicity vs. asychronicity). The study received ethical approval by the Università della Svizzera italiana Institutional Review Board (approval date 30/10/2018).

### Development and evaluation of experimental material

Before conducting the main experiment, we performed a pilot study to develop and evaluate the experimental material to the specific context of Canton Ticino (Italian-speaking Switzerland), where the study was conducted. The pilot study focused on the identification of which favors and responses could best fit the balance between a scientific setting and the ecological component of the proposed interactions. Participants in the pilot study were 107 young adults (Mage = 26.05, SD = 3.97; 67.3% female). A first set of twelve different favors was defined. Every favor was evaluated with respect to plausibility ("how realistic is the requested favor to you?"), frequency ("how many times did you find yourself in this situation, as the person in need or the one providing help?"), time and effort required to satisfy the request ("in your opinion, how much time/effort is required in order to satisfy this favor?"). A 5-points Likert scale was employed to rate the four dimensions, ranging from 1 = "Not at all/Never" to 5 = "Totally/A lot". The selection criteria were: 1) higher summed score of the means across the above-mentioned dimensions (plausibility, frequency, time, and effort), where plausibility was double-weighted to preserve the ecological validity, and 2) differentiation to avoid similar requests (e.g., mother tongue lessons before an interview vs. adjustment of CV). Three favors were finally selected: (1) ask Sophie/Mark to offer lessons in her/his mother tongue before an interview, (2) help with a work/study project, and (3) help moving a couch.

Six empathic and six non-empathic responses were evaluated from the most negative to the most positive on a 100-point visual analogue scale, ranging from -50 = "Very negative" to +50 = "Very positive". The highest three for each polarity were adapted to the help-related contexts and embedded in the experiment. The detailed results of the pilot study are available upon request.

A set of popular emojis, equally split between positive and negative, was selected by using the emojis tracker for Twitter (https://www.emojitracker.com/), as done in previous studies [81, 82]. For empathic emojis, the following ones were selected: "Smiling face with smiling eyes", "Smiling face with sunglasses", and "Smiling face with open mouth and smiling eyes", while both "Angry faces" (both yellow and red) and the "Unamused face" were chosen for the non-empathic responses. S1 Fig includes examples of the manipulated messages.

### Experimental study procedure

Participants were recruited through Facebook posts, flyers, and snowball sampling. The study focused on young adults, as they are particularly active on WhatsApp [11]. Only participants between 18 and 35 years of age, fluent in Italian, were eligible for the experimental study. A quota sampling strategy was applied to assure a balanced distribution of gender. Prior to the start of the experimental session, participants received a brief introduction addressing potential ethical concerns, and they signed the informed consent.

First, participants had to complete a questionnaire assessing socio-demographic characteristics. Next, participants received a smartphone, with printed information, describing the previously developed gender-matched scenario: they were new in the city and got the number of Mark (for male participants) or Sophie (for female participants). The participants received the instruction to contact the peer through WhatsApp (installed on the smartphone device) and ask him/her three favors from a list provided by the researchers. The necessary contact details

were already saved in the smartphone. After participants briefly introduced themselves ("Hi Sophie/Mark, I am [participant's name]. We exchanged our phone numbers a few days ago"), they asked the first of three favors. To increase the ecological validity of the experiment, they could choose the order of the favors. The peer (Mark or Sophie) was a researcher located in another room using WhatsApp (on a dual SIM smartphone) to simulate the male character (with the associated phone number) or the female character (with the associated phone number). Based on the randomly assigned condition, the participants received one of eight manipulated answer options (see S1 Fig for examples).

Between each of the three favor-answer-interaction, participants saw a 2-minutes video with images of peaceful landscapes that functioned as a neutral stimulus. After the experimental phase with the three favor-answer-interaction was over, the participants received a final questionnaire assessing post-test variables. At the end of the experiment, they received a short debriefing and 30 Swiss Francs for compensation. The flowchart in S2 Fig summarizes the experimental study procedure.

## Experimental study measures

All measures were self-report and, if available, validated scales were used. Forward and backward translations from English to Italian and *vice versa* by two independent translators were implied to assure linguistic validity. If necessary, scales were adapted to the adult context (e.g., items referring to the school environment were adapted to college and/or work situations). Measures were assessed via an online questionnaire in Qualtrics, accessible on a computer in the laboratory where the experiment was carried out. Within a scale, the item order was randomized, whereas the order of the scales was predetermined.

**Perceived social presence.** The perceived component of social presence was assessed with a set of semantic differentials based on previous literature [83]. Eight bipolarized adjectives were chosen to address the social presence of the peer. Participants were asked to evaluate the peer as being: 1) sensitive–insensitive, 2) warm–cold, 3) responsive–non-responsive, 4) interactive–non-interactive, 5) near–distant, 6) personal–impersonal, 7) authentic–superficial, and 8) sociable–non-sociable. Answer options ranged from 1 to 7. All scores were averaged, with higher scores reflecting a higher level of perceived social presence. Cronbach's Alpha was good, $\alpha = .87$, as well as the inter-item correlations, $r > .47$. Confirmatory factor analysis (CFA) revealed that the one-factor tested model reached good fit of the data, $\chi^2 (19) = 35.634$, $p = .012$, SRMR = .0434, RMSEA = .074, CFI = .978.

**Subjective social presence.** The self-report measure developed by Burgoon and Hale [84] was applied to assess subjective social presence. A total of eleven items, belonging to Similarity/Receptivity/Inclusion, Non-Immediacy, and Formality subscales, were employed and adapted to the context of the experiment. Measured on a Likert scale, response options ranged from 1 = "Strongly agree" to 7 = "Strongly disagree". Example items are: "Mark/Sophie tried to establish a relationship" (Similarity/Receptivity/Inclusion subscale), "Mark/Sophie was very detached" (non-immediacy subscale), and "Mark/Sophie was not responsive to my ideas" (formality subscale). A total score was obtained by averaging single item scores, with higher scores corresponding to higher levels of subjective social presence. Cronbach's Alpha was good, $\alpha = .89$, as well as the inter-item correlations, $r > .41$. CFA revealed that the one-factor tested model reached good fit of the data, $\chi^2 (134) = 220.960$, $p < .001$, SRMR = .0566, RMSEA = .064, CFI = .953.

**Interpersonal trust.** The original Trust in Close Relationships scale [49] was employed to assess interpersonal trust in the dyadic communicative exchange. Due to the loving-related and intimate nature of the predictability dimension, the related items referred to this domain

were removed. A total of 18 items, belonging to Faith and Dependability original subscales, were adapted to the experimental context by weakening the intimate texture of the dimensions. Measured on a Likert scale, response options ranged from 1 = "Strongly disagree" to 7 = "Strongly agree". Example items are: "I can rely on Mark/Sophie to react in a positive way when I expose my weaknesses to him/her" (Faith subscale) and "Mark/Sophie has proven to be trustworthy and I am willing to let him/her engage in activities which other partners find too threatening" (Dependability subscale). A total score was obtained by averaging single item scores, with higher scores corresponding to higher levels of trust in Mark/Sophie. Cronbach's Alpha was good, $\alpha = .92$, as well as the inter-item correlations, $r > .40$. CFA revealed that the one-factor tested model reached good fit of the data, $\chi^2 (134) = 220.960$, $p < .001$, SRMR = .0566, RMSEA = .064, CFI = .953.

**Social support.** Received social support was assessed with the Actually Received Support scale of the Berlin Social Support Scale (BSSS; [85]). The scale structure is oriented on the dyadic supportive behavior and how it is perceived. Given the experimental study design, which was intended to create a needing-help-environment anchored, among others, on time and effort required to satisfy the requests, the informational subscale was excluded. Therefore, the scale was composed by 13 items, referring to emotional support (EMO; 9 items, example item: "Mark/Sophie did not show much empathy for my situation"), instrumental support (INST; 3 items, example item: "Mark/Sophie took care of things I could not manage on my own") and the final item of satisfaction with support (SAT; "In general, I am very satisfied with the way Mark/Sophie behaved"). Measured on a Likert scale, response options ranged from 1 = "Strongly disagree" to 4 = "Strongly agree". A total score was obtained by averaging responses, where higher score indicated higher levels of perceived support received from Sophie/Mark. Cronbach's Alpha was good, $\alpha = .97$, as well as the inter-item correlations, $r > .70$. CFA revealed that the one-factor tested model reached good fit of the data, $\chi^2 (50) = 96.367$, $p < .001$, SRMR = .0260, RMSEA = .063, CFI = .978.

**Personality attribution.** Perception of marker personality was assessed using three items, capturing the Ashton, Lee and Goldberg [86] HEXACO model of personality. The model foresees six personality dimensions: Honesty/Humility, Emotional stability, Extraversion, Agreeableness, Conscientiousness and Openness. For the purpose of the present research, Agreeableness, Extraversion, and Emotional stability were chosen as interpersonal traits [76]. Participants were asked to report how much they would describe Mark/Sophie as 1) calm, 2) agreeable, and 3) outgoing. Each adjective was rated on a 7-point Likert scale, with answer options ranging from 1 = "Strongly disagree" to 7 = "Strongly agree". Cronbach's Alpha was good, $\alpha = .79$, as well as the inter-item correlations, $r > .56$. Higher scores indicated higher levels of the respective personality dimension.

## Experimental study participants

A total of 160 young adults took part in the experimental study (Mage = 23.38, SD = 3.99, range 18–35 years old). These participants were different from the sample of the pilot study to evaluate the experimental study material. Participants were balanced in gender (50%, N = 80, male) and they were primarily based in Ticino, Switzerland (88.8%. N = 142), high school graduated (40%, N = 64) or college graduated (38.8%, N = 62). College students represented 65.6% (N = 105) of the sample.

## Experimental data analysis

Because a forced-answer format was used for all measures, the data did not include any missing values. Descriptive statistics, bivariate correlations, and regression analyses were

performed in SPSS version 25 [87]. The hypothesized model was tested using the lavaan package [88] in R software [89], using maximum likelihood estimation with robust standard errors and a Satorra-Bentler scaled test statistic. The three features of SMC were dummy coded as following: synchronous (1) vs asynchronous (0), modality with emojis (1) vs modality without (0), and positive emphatic valence (1) vs negative non-empathic valence (0).

## Results

Correlations between all measures are shown in Table 1. Subjective social presence, perceived social presence, social support, interpersonal trust, and personality attribution were all positively and significantly correlated to each other. The synchronicity and the modality of the message correlated significantly with perceived social presence, whereas the valence of the message correlated significantly with all the other variables.

The hypothesized model (see Fig 1) reached good fit of the data, $\chi^2$ (6) = 8.24, p = .22, CFI = .99, SRMR = .012, RMSEA = .048, 90% CI [.000, .119]. Examination of path coefficients yielded evidence for the hypothesized paths, with few exceptions. Fig 2 represents the standardized regression weights of the constructs included in the model.

Social support was predicted by both the subjective ($\beta$ = .14, p < .01) and perceived social presence ($\beta$ = .16, p < .01), showing that an increase in social presence corresponds to an increase in social support. Moreover, personality attribution to Sophie/Mark was positively predicted by subjective social presence ($\beta$ = .43, p < .01) but not by perceived social presence. The two social presence variables were not significantly related to interpersonal trust. Furthermore, subjective and perceived social presence significantly correlated with each other (r = .624, p < .01). Message valence was significantly related to both subjective ($\beta$ = .78, p < .01) and perceived social presence ($\beta$ = .62, p < .01), indicating that when the valence of the message was positive (empathic) social presence increased. Likewise, modality was positively related to both subjective ($\beta$ = .17, p < .01) and perceived social presence ($\beta$ = .16, p < .01). Furthermore, synchronicity was positively associated with perceived social presence ($\beta$ = .16, p < .01), but not with subjective social presence.

Moreover, the indirect paths involving subjective social presence as a mediator were significant between valence and social support ($\beta$ = .11, p < .05) as well as valence and personality ($\beta$ = .33, p < .01), and modality and personality ($\beta$ = .07, p < .01). On the other side, the indirect only significant paths involving perceived social presence as a mediator were the ones between synchronicity and social support ($\beta$ = .03, p = .03), modality and social support ($\beta$ = .03, p = .02), and, once again, valence and social support ($\beta$ = .10, p < .01). No significant indirect effects were found for personality attribution (see Table 2 for full details of the model).

**Table 1. Bivariate correlations, means and standard deviations.**

|  | M (SD) | 1 | 2 | 3 | 4 |
|---|---|---|---|---|---|
| Subjective Social Presence | 3.43 (1.46) | .62*** | .43*** | .27*** | .49*** |
| 1 Perceived Social Presence | 3.37 (1.39) |  | .51*** | .26*** | .34*** |
| 2 Social Support | 2.15 (0.98) |  |  | .55*** | .47*** |
| 3 Interpersonal Trust | 3.07 (1.18) |  |  |  | .44*** |
| 4 Personality Attribution | 3.63 (1.58) |  |  |  |  |

Partial correlation coefficients controlled for experimental condition;

* p < .05;

*** p < .001.

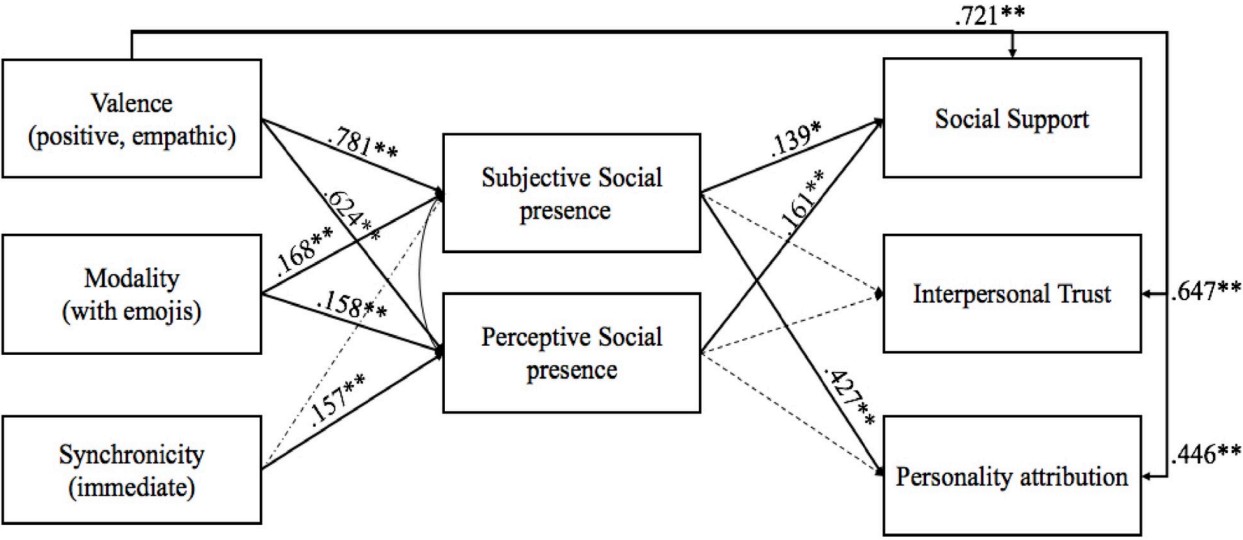

**Fig 2. Final model with social support, interpersonal trust, and personality attribution as dependent variables and subjective and perceived social presence as mediators.** Standardized coefficients are shown. Reference groups: valence (negative, non-empathic), modality (without emojis), synchronicity (7 minutes delay). Dashed arrows denote non-significant paths; all other paths denote significant relationships at * p < .05, ** p < .01, *** p < .001.

## Discussion

The portability of the smartphone, the perception of its indispensability for everyday activities [13], and its predominant use in the context of social interactions and interpersonal communication [14] make smartphone-mediated communication (SMC) and its effect on interpersonal functioning a crucial point to be studied.

The present study tested a partial mediation model considering SMC features (i.e., synchronicity, modality, and valence of the message) as predictor, social presence (subjective and perceived) as mediator, and social support, interpersonal trust, and the evaluation of the communication partner's personality as outcome variables. SMC features were manipulated in a randomized experiment and mediator and outcome variables were measured

**Table 2. Path coefficients and significance level of the final Structural Equation Model.**

| | Mediators | | | | | | Outcomes | | | | | | | | |
|---|---|---|---|---|---|---|---|---|---|---|---|---|---|---|---|
| | Social presence Perceptive | | | Interpersonal trust | | | Social presence Perceptive | | | Interpersonal trust | | | Social presence Perceptive | | |
| **Experimental Conditions** | B | SE | β | B | SE | β | B | SE | β | B | SE | β | B | SE | β |
| Modality (with emojis) | 0.49 | 0.14 | .17*** | 0.44 | 0.16 | .16*** | | | | | | | | | |
| Synchronicity (synchronous) | -0.09 | 0.14 | -.03 | 0.43 | 0.16 | .16*** | | | | | | | | | |
| Valence (empathic) | 2.27 | 0.14 | .78*** | 1.73 | 0.16 | .62*** | 1.52 | 0.19 | .65*** | 1.41 | 0.11 | .72*** | 1.40 | 0.20 | .45*** |
| **Mediators** | | | | | | | | | | | | | | | |
| Subjective social presence | | | | | | | 0.13 | 0.08 | .16 | 0.10 | 0.04 | .14** | 0.46 | 0.08 | .43*** |
| Perceived social presence | | | | | | | 0.09 | 0.05 | .10 | 0.11 | 0.03 | .16*** | 0.06 | 0.07 | .06 |

** p < .01;

*** p < .001.

immediately after the experiment. Based on data from 160 young adults, we found that message valence and modality were both associated with social presence, measured with its perceptual and subjective components. On average, empathic messages and the use of emojis increased the perception of the communication partner as real and present in the SMC conversation. These results confirm findings from previous literature. Indeed, empathic accuracy has been found to increment social presence in mediated communication [41, 42] the elicitation of positive emotions [45] and satisfaction with the relationship [43]. Furthermore, the use of verbal and visual cues has been previously shown to influence the positive experience of social presence [7]. According to the principles of Media Richness Theory, emojis can enhance the message value by mimicking the natural non- and para-verbal language of the communicator [20–22]. That said, the use of emojis in the present study reinforced the meaning of the message and compensated for the lack of non-verbal and para-verbal cues, thus enhancing the emotional value of the message [28], which eventually led to higher levels of social presence [30].

Previous literature stated that long pauses in the mediated communication process decrease perceptions of "being there" as well as the awareness and imagination of the other person [27]. In a similar way, our results indicate that the synchronicity of the message (i.e., without a temporal delay) increases the perceptual component of social presence only, which is the evaluation of the other's accessibility and the extent of how distant the other is perceived. However, the synchronicity of the message does not affect the subjective social presence, which is the perception of being concordant and symmetric with the other. Concordance relates to agreement, harmony, and responsiveness, which are strongly connected to the message valence and not to the perceived immediacy of the SMC, although non-immediacy was a subscale measured as part of subjective social presence. It seems that the latter is overruled by the other measures of subjective social presence to capture the perception of being concordant with the communication partner. A second explanation for this finding may be that the suggested scenarios were built on different requests for help, which created unbalance in the symmetry in the communication partners.

Finally, it was expected that social presence partially mediates the relations between SMC and interpersonal trust, social support, and personality attribution. Our results show that subjective social presence, indeed, significantly increases the social support received from the communication partner, and it leads to a more positive evaluation of his/her personality. Higher levels of social support are also predicted by perceived social presence. Furthermore, similarly to face-to-face communication, our results demonstrate that SMC allows the formation of first impressions about others' personalities [66] based on their online behaviors [67, 68]. The formation of first impressions is context-dependent and highly vulnerable to patterns and biases [74], especially in computer/smartphone-mediated communication [90], and it depends on the perception of being concordant with the other.

On the other hand, interpersonal trust is not linked to either subjective or perceived social presence, but it is directly predicted by the valence of the message. Thus, contrary to what emerged in communication research in other online environments (i.e., B2C e-Commerce; [55, 56]), during a conversation that imitates the everyday interactions among peers, it is the empathic vs. non-empathic content of the message that influences the perception of trust. Neither modality nor synchronicity had an impact on trust. Thus, interpersonal trust, as a combination of faith and dependability, may be released from the perception of the other as emotionally concordant and be more related to the ready availability of the other.

## Limitations

Despite the experimental design, which allows to make causal claims about the impact of SCM on social presence and the quality of the constructed interpersonal relationship, the present research has several limitations. First, the small sample size does not allow to consider all the interactions between the three experimental conditions in the full model. It would be interesting to analyze whether social presence, and, subsequently, social support, trust, and personality attribution would change according to the combination of synchronicity, modality, and message valence. Second, as for every other experimental study, the ecological validity is limited. Although our pilot study helped to identify scenarios, favors, and emojis young adults can relate to, other aspects could threat the ecological validity. For example, the 7-minutes delay in the asynchronous condition may be somewhat arbitrary and not perceived as asynchronous in real-life SMC interactions. Third, since the measures are all self-reported, we cannot exclude a degree of social desirability. Future research should investigate SMC applying self-report measures in combination with objective measures such as psycho-physiological correlates (e.g., heart rate, electrodermal activity) as done in other contexts of communication and media research (e.g., [91–93]).

## Conclusions

The results of the present experimental study provide important answers related to the immediate impact of SMC on people's personal feelings and subjective experience while interacting with others. In sum, the results underline that synchronous empathic messages with emojis are valid predictors of social presence, which, in turn, is positively linked to social support and personality attribution. Moreover, the attribution of trust to a peer, social support, and personality attribution are directly determined by the empathic accuracy of the message, i.e. when the response matches the request by the individual required support. A proverb says: "It's not what you say, it's how you say it." In our study on SMC, however, it should be rewritten to underline: "It is both what you say and how you say it that makes the difference".

This conclusion stresses the importance of learning about the implications of our everyday SMC practices, ranging from informal family and peer-to-peer communication to SMC in workplace settings or customer care. Given that SMC already starts in preadolescence, where smartphone ownership has gone up to 90 percent among 11 year-olds [94], learning about the implications on the perceptions and feelings induced by different ways of SMC should be on the agenda both at home but also in school curricula aimed at promoting a conscious use of the smartphone. Such a use is not only beneficial for subjective well-being but also for the initiation and maintenance of good interpersonal relationships.

Besides these practical implications, the present study lays the ground for future experimental research on SMC. In particular, future studies should include experimental observations in natural settings, over time, pay more attention to smartphone addictive behaviors, and consider younger age populations. Since the present study allows to conclude on the short-term impact of SMC only, this study should be a starting point for future research on the long-term effects in natural settings. Furthermore, the proposed tasks in the present study allow only one reaction by the participant but no turn taking as it is often the case in SMC. Thus, to augment ecological validity, future research should extend the duration of SMC and participants to respond to a manipulated response. Thematic analysis of such responses would then allow to deduct different forms of reactions that are manifestations of perceptions of social presence, trust, and support, investigated in the present study.

Another idea for future research is the extension of the current project to many-to-many SMC as it has been done in the longitudinal study by Walther [95] on computer-mediated

group interactions to analyze different relational communication dimensions. Transferred to the context of SMC, it would be interesting to investigate synchronicity, modality, and message valence used by participants during a group chat on different topics initiated by the researcher over the course of several weeks. The use of ecological momentary assessment evaluations [96] on emotional state and social presence would allow to collect detailed information on the subjective effects of the interaction without taking participants out of their everyday environment.

## Supporting information

**S1 Fig. Experimental material for both synchronous (immediate) and asynchronous (7 minutes delay) conditions: a) = with emojis, positive answers, b) text only, positive answers, c) with emojis, negative answers, d) text only, negative answers.**
(DOCX)

**S2 Fig. Step-by-step procedure implemented in the study.** Order of the favors ("language lessons", "work/study project", and "moving couch") randomly chosen by the participants.
(DOCX)

**S1 File. Digital lives Totale dum.**
(SAV)

**S2 File. Modello finale.**
(R)

## Acknowledgments

We would like to thank Chiara Antonietti and Chiara Filipponi for their help during data collection.

## Author Contributions

**Conceptualization:** Serena Petrocchi, Laura Marciano, Anne-Linda Camerini.

**Data curation:** Anna Maria Annoni.

**Formal analysis:** Laura Marciano, Anna Maria Annoni.

**Funding acquisition:** Serena Petrocchi, Anne-Linda Camerini.

**Project administration:** Serena Petrocchi, Anna Maria Annoni, Anne-Linda Camerini.

**Supervision:** Serena Petrocchi, Anne-Linda Camerini.

**Writing – original draft:** Serena Petrocchi, Anna Maria Annoni.

**Writing – review & editing:** Serena Petrocchi, Laura Marciano, Anne-Linda Camerini.

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
