## [Decision Letter · Decision Letter 0]

17 Jun 2020

PONE-D-20-10424

"What you say and how you say it” matters:  An experimental evidence of the role of synchronicity, modality, and message valence during smartphone-mediated communication

PLOS ONE

Dear Dr. Petrocchi,

Thank you for submitting your manuscript to PLOS ONE. After careful consideration, we feel that it has merit but does not fully meet PLOS ONE’s publication criteria as it currently stands. Therefore, we invite you to submit a revised version of the manuscript that addresses the points raised during the review process.

We look forward to receiving your revised manuscript.

Kind regards,

Frantisek Sudzina

Academic Editor

PLOS ONE

2. We noted in your submission details that a portion of your manuscript may have been presented or published elsewhere.

"A preliminary version of the manuscript has been submitted to the 70th annual conference of the International Communication Association in 2020."

Please clarify whether this conference proceeding was peer-reviewed and formally published. If this work was previously peer-reviewed and published, in the cover letter please provide the reason that this work does not constitute dual publication and should be included in the current manuscript.

Reviewers' comments:

Reviewer's Responses to Questions

**Comments to the Author**

1. Is the manuscript technically sound, and do the data support the conclusions?

Reviewer #1: Yes

Reviewer #2: Yes

2. Has the statistical analysis been performed appropriately and rigorously? 

Reviewer #1: Yes

Reviewer #2: Yes

3. Have the authors made all data underlying the findings in their manuscript fully available?

Reviewer #1: Yes

Reviewer #2: Yes

4. Is the manuscript presented in an intelligible fashion and written in standard English?

Reviewer #1: Yes

Reviewer #2: Yes

5. Review Comments to the Author

Reviewer #1: Manuscript title: "What you say and how you say it” matters: An experimental evidence of the role of synchronicity, modality, and message valence during smartphone-mediated communication

MINOR revisions

Summary:

I would like to congratulate the authors because the manuscript is really interesting and well written. It reports a well-structured mediation analysis concerning Smartphone-Mediated Communication across a sample of young individuals. This work should have a relevant impact on literature. However, in my humble opinion, some points need further attention to improve the value of the article.

Major comments:

Experimental Design:

Line 192: The Authors declare a 2X2X2 experimental design. However, it is not clear which are the conditions of the factorial design are – and how they were manipulated.

Line 197: In addition, the Authors state: «the participants communicated with a peer»: Sophie (for females) or Mark (for males). However, the authors should explain why they did not choose for a peer sex balance which the participants communicated with.

Line 204: the Authors introduce a “Pilot study”. Unfortunately, it’s not completely clear if both the procedure (line 235), measures (262) refers to the “Main study” or the “pilot study” – probably because there are no results reported from this “pilot study” (but are available on request). Thus, probably it could be helpful to provide some results of this pilot study.

Data Analysis:

Table 1: some correlation values suggest the presence of multicollinearity (e.g.: variable#7 and variable#2). Is it correct? Moreover, it’s not completely clear why correlations between variable#5, #6, and #7 are not applicable (n/a).

Discussion:

The discussion section should be implemented by a deepened analysis of the results across the theoretical framework.

Moreover, the discussion section should be adjusted considering previous comments.

Minor comments:

Line 6: it is not immediately understandable what SNC refers to. The first time a concept is introduced it should be avoided to use only its acronym.

Line 67-73: please, add at least one reference.

Line 83-90: please, add at least one reference.

Line 135-139: please, add at least one reference.

Table 2: table 2 is not easy to interpret and could be better structured.

Figures 1 and 2: both figures are too small and not easy to read – and they don't do justice to such beautiful research work. Could it be possible to propose a more suitable graphic version?

Reviewer #2: The present experimental study aimed at investigating the impact of three aspects of SMC, i.e. synchronicity, modality, and valence, on interpersonal trust, social support, and personality attribution, through the mediation of perceived and subjective social presence of the communication partner.

The approach used for experimental manipulation is innovative and of scientific interest.

However, I suggest strengthening some aspects, above all of a theoretical nature, in order to make the study ready for publication

1. In the abstract I would spend a few more words to explain the constructs and variables (not in detail), while I would reduce the first sentence which is not specifically related to this study “Although past research has investigated the differential effects of online and offline communication on psychosocial functioning, smartphone-mediated-communication and instant messaging are fast-changing environments in which technological advancements spread quickly and widely in the population.”

2. Also in the introduction I would cut the part about the historical evolution of technologies while I would explain the importance of this study. Why is it important to study the relationship between the variables investigated?

From the premises to the goal of the study there is a jump (“This experimental study aims to contribute to scientific evidence on the role of specific aspects of SMC (i.e. synchronicity, modality, and message valence) in the perception of the communication partner (i.e. social presence) and the quality of the interpersonal relationship in the context of SMC. Before introducing the study, the scientific literature on SMC and related concepts will be reviewed to derive a theoretical model guiding the experimental study and the interpretation of its results.”). I suggest to better focus the variables that will be manipulated and tested before stating the goal of the study

3. Please better explain how did you manipulate the empathic accuracy. At page 13 you stated “Empathic accuracy is defined as the degree to which a person infers what other people think and feel [38,39], enhancing the experience of positive or negative feelings and emotions.” I do not understand why positive valence is considered an expression of empathic accuracy (vs. negative valence, i.e. empathic inaccuracy)

4. I suggest to include a figure representing the experimental design to facilitate readers in understanding the procedure and study design

5. the “pilot” study seems more a “preliminary” study. This preliminary study should be introduced before in the text. Did you involve the same participants?

6. Discussion. The relevance of the study and implications should be enhanced

6. PLOS authors have the option to publish the peer review history of their article (what does this mean?). If published, this will include your full peer review and any attached files.

Reviewer #1: No

Reviewer #2: Yes: Daniela Villani

---

## [Author Response · Author response to Decision Letter 0]

28 Jul 2020

Dear Editorial Team, 

Dear Reviewers, 

Please find below our answers to your suggestions/revisions. 

All my best

Serena Petrocchi

**

Reviewers' comments:

Reviewer's Responses to Questions

Comments to the Author

1. Is the manuscript technically sound, and do the data support the conclusions?

Reviewer #1: Yes

Reviewer #2: Yes

2. Has the statistical analysis been performed appropriately and rigorously?

Reviewer #1: Yes

Reviewer #2: Yes

3. Have the authors made all data underlying the findings in their manuscript fully available?

Reviewer #1: Yes

Reviewer #2: Yes

4. Is the manuscript presented in an intelligible fashion and written in standard English?

Reviewer #1: Yes

Reviewer #2: Yes

5. Review Comments to the Author

Reviewer #1: Manuscript title: "What you say and how you say it” matters: An experimental evidence of the role of synchronicity, modality, and message valence during smartphone-mediated communication

MINOR revisions

Summary:

I would like to congratulate the authors because the manuscript is really interesting and well written. It reports a well-structured mediation analysis concerning Smartphone-Mediated Communication across a sample of young individuals. This work should have a relevant impact on literature. However, in my humble opinion, some points need further attention to improve the value of the article.

Major comments:

Experimental Design:

Line 192: The Authors declare a 2X2X2 experimental design. However, it is not clear which are the conditions of the factorial design are – and how they were manipulated.

Authors: thank you for your comment. The three experimental conditions are the valence of the message (empathic accurate vs. empathic inaccurate message), the modality (with or without emojis), and the synchronicity of the message (immediately after the participant sent the request vs. 7 minutes later). To increase statistical power, we decided to collapse the conditions to be able to perform a SEM analysis and test for the hypothesized mediation of social presence. That said, we understand that the sentence can be misleading and we deleted it from the revised version of the paper. 

Line 197: In addition, the Authors state: «the participants communicated with a peer»: Sophie (for females) or Mark (for males). However, the authors should explain why they did not choose for a peer sex balance which the participants communicated with.

Authors: we decided to match the gender of the peer to the gender of the participant for two main reasons. First, there is evidence showing that females and males have different communication styles during mediated communication (e.g., Savicki, Kelley, 2004) and different aims during their interactions via smartphone (e.g., Park, Lee, 2014). For this reasons, we decided to match peer’s gender and participant’s gender. If we would not have matched the gender, we should have had to control for the influence of interacting with a same-gender peer or a different-gender peer. This would have required a larger sample that, given the available resources, we could not reach. We implemented this consideration into the manuscript. Second, we wanted to avoid any romantic connotations. You can already find this consideration in the manuscript (see line 197). 

Line 204: the Authors introduce a “Pilot study”. Unfortunately, it’s not completely clear if both the procedure (line 235), measures (262) refers to the “Main study” or the “pilot study” – probably because there are no results reported from this “pilot study” (but are available on request). Thus, probably it could be helpful to provide some results of this pilot study.

Authors: the part of the text from line 211 to line 245 pertains to the pilot study with the information that detailed results of the pilot study are available upon request (we added “detailed” in the manuscript, line 232). The part from line 247 to line 397 pertains to the experimental study. We clarified this point now including a specification in the titles of the Procedure and Measures sections and a specification in line 345 regarding study participants. 

Data Analysis:

Table 1: some correlation values suggest the presence of multicollinearity (e.g.: variable#7 and variable#2). Is it correct? Moreover, it’s not completely clear why correlations between variable#5, #6, and #7 are not applicable (n/a).

Authors: many thanks for this important observation. In fact, the presence of multicollinearity depended on the shared variance due to the three experimental manipulation variables that added a common variance to the correlations between the variables. We calculated a new table of partial correlations controlling for the three conditions (i.e., valence, modality, and synchronicity). As you can see now, the correlations are lower than before suggesting an association between construct but not an overlap or signs of multicollinearity. 

The n/a was added because the correlations between the three experimental conditions are meaningless, so in this sense “not applicable”. We acknowledge that including these three variables in the table could mislead a reader. Thus, we decided to delete them from the table. 

Discussion:

The discussion section should be implemented by a deepened analysis of the results across the theoretical framework.

Moreover, the discussion section should be adjusted considering previous comments.

Authors: we checked and adjusted our discussion based on the revisions of the previous sections of our manuscript. Based on the suggestions by Reviewer 2, we have also expanded on the practical and scientific implications of our study in the conclusions section.

Minor comments:

Line 6: it is not immediately understandable what SNC refers to. The first time a concept is introduced it should be avoided to use only its acronym.

Authors: we have specified the acronym in the abstract where first time it is mentioned. 

Line 67-73: please, add at least one reference.

Authors: we did, thank you

Line 83-90: please, add at least one reference.

Authors: we did, thank you

Line 135-139: please, add at least one reference.

Authors: we did, thank you

Table 2: table 2 is not easy to interpret and could be better structured.

Authors: thank you. We slightly changed the layout of the table and hope that it is now easier to read interpret.

Figures 1 and 2: both figures are too small and not easy to read – and they don't do justice to such beautiful research work. Could it be possible to propose a more suitable graphic version?

Authors: thank you for your comment, we enlarged the figures. 

Reviewer #2: The present experimental study aimed at investigating the impact of three aspects of SMC, i.e. synchronicity, modality, and valence, on interpersonal trust, social support, and personality attribution, through the mediation of perceived and subjective social presence of the communication partner.

The approach used for experimental manipulation is innovative and of scientific interest.

However, I suggest strengthening some aspects, above all of a theoretical nature, in order to make the study ready for publication

1. In the abstract I would spend a few more words to explain the constructs and variables (not in detail), while I would reduce the first sentence which is not specifically related to this study “Although past research has investigated the differential effects of online and offline communication on psychosocial functioning, smartphone-mediated-communication and instant messaging are fast-changing environments in which technological advancements spread quickly and widely in the population.”

Authors: thank you for your comment. We have deleted the first sentence and explained in more details what are the three experimental conditions, the mediators, and the expected outcomes. 

2. Also in the introduction I would cut the part about the historical evolution of technologies while I would explain the importance of this study. Why is it important to study the relationship between the variables investigated? From the premises to the goal of the study there is a jump (“This experimental study aims to contribute to scientific evidence on the role of specific aspects of SMC (i.e. synchronicity, modality, and message valence) in the perception of the communication partner (i.e. social presence) and the quality of the interpersonal relationship in the context of SMC. 

Before introducing the study, the scientific literature on SMC and related concepts will be reviewed to derive a theoretical model guiding the experimental study and the interpretation of its results.”). I suggest to better focus the variables that will be manipulated and tested before stating the goal of the study

Authors: we cut the historical evolution of technologies leaving only the part that really matters with our study and is important as introduction to the study of SMC. We better explained what is the contribution of our study and introduced the variables under analysis. 

3. Please better explain how did you manipulate the empathic accuracy. At page 13 you stated “Empathic accuracy is defined as the degree to which a person infers what other people think and feel [38,39], enhancing the experience of positive or negative feelings and emotions.” I do not understand why positive valence is considered an expression of empathic accuracy (vs. negative valence, i.e. empathic inaccuracy)

Authors: please see page 13. We have better defined what empathic accuracy means considering the literature and the context of our research. Empathic accuracy is defined as the degree to which a person infers what other people think and feel [38,39], enhancing the experience of positive or negative feelings and emotions. In this vein, an empathic accurate answer to a request is a message in which the sender takes into consideration the receiver’s mental state (i.e., what he/she thinks and how he/she feels) and replies accordingly, thus creating a positive experience. Based on this definition, an answer that takes into account the thoughts and feelings of the receiver is an empathic accurate message and has a positive valence. Contrarily, an answer that does not take into account the thoughts and feelings of the receiver is an empathic inaccurate message and has a negative valence.

4. I suggest to include a figure representing the experimental design to facilitate readers in understanding the procedure and study design

Authors: please find the flowchart we have created. 

5. the “pilot” study seems more a “preliminary” study. This preliminary study should be introduced before in the text. Did you involve the same participants?

Authors: the pilot study involved different participants than those that have been involved in the main experimental study. The main aim of the pilot study was to develop and evaluate the experimental material, i.e. the scenarios and the responses in a simulated context of smartphone-mediated communication to be able to test our research hypotheses. More precisely, in the pilot study, we wanted to understand which scenarios were the most plausible, frequent, and realistic for our social context given the fact that we adapted an experimental procedure developed by Feng et al (2004). Moreover, we wanted to test whether the developed responses were perceived as empathic (or non-empathic). As specified in the manuscript, we approached a separate sample for this pilot study. We specified this aspect now in the manuscript in the paragraph titles “Experimental Study Participants”. Since the pilot has different participants than the main study but has research questions related to the main study, we refer to it as a pilot study instead of preliminary study. 

6. Discussion. The relevance of the study and implications should be enhanced

 Authors: thank you for your comment. We revised the discussion to highlight better the implications of our research and its contribution to further develop research in the field of smartphone-mediated communication. 

6. PLOS authors have the option to publish the peer review history of their article (what does this mean?). If published, this will include your full peer review and any attached files.

Do you want your identity to be public for this peer review? For information about this choice, including consent withdrawal, please see our Privacy Policy.

Reviewer #1: No

Reviewer #2: Yes: Daniela Villani

---

## [Editor Report · Decision Letter 1]

5 Aug 2020

"What you say and how you say it” matters:  An experimental evidence of the role of synchronicity, modality, and message valence during smartphone-mediated communication

PONE-D-20-10424R1

Dear Dr. Petrocchi,

We’re pleased to inform you that your manuscript has been judged scientifically suitable for publication and will be formally accepted for publication once it meets all outstanding technical requirements.

Kind regards,

Frantisek Sudzina

Academic Editor

PLOS ONE

---

## [Editor Report · Acceptance letter]

8 Sep 2020

PONE-D-20-10424R1 

“What you say and how you say it” matters: An experimental evidence of the role of synchronicity, modality, and message valence during smartphone-mediated communication 

Dear Dr. Petrocchi:

I'm pleased to inform you that your manuscript has been deemed suitable for publication in PLOS ONE. Congratulations! Your manuscript is now with our production department. 

Kind regards, 

on behalf of

Dr. Frantisek Sudzina 

Academic Editor

PLOS ONE